# Generation and Superposition of Perfect Vortex Beams in Terahertz Region via Single-Layer All-Dielectric Metasurface

**DOI:** 10.3390/nano12173010

**Published:** 2022-08-30

**Authors:** Qi Wu, Wenhui Fan, Chong Qin

**Affiliations:** 1State Key Laboratory of Transient Optics and Photonics, Xi’an Institute of Optics and Precision Mechanics, Chinese Academy of Sciences, Xi’an 710119, China; 2School of Opto-Electronic Technology, University of Chinese Academy of Sciences, Beijing 100049, China; 3Collaborative Innovation Center of Extreme Optics, Shanxi University, Taiyuan 030006, China

**Keywords:** all-dielectric metasurface, perfect vortex beam, perfect Poincaré beam, terahertz

## Abstract

Terahertz (THz) orbital angular momentum (OAM) technology provides promising applications in future wireless communication with large bandwidth and high capacity. However, the ring radius of the conventional THz vortex beam is related to the topological charge, limiting the co-propagation of multiple OAM modes in the THz communication systems. Although the perfect vortex beam (PVB) based on traditional methods can solve this problem, they are usually bulky and unstable. Here, we demonstrate two PVB generators based on a single all-dielectric metasurface to obtain polarization-independent PVB and spin multiplexed PVB, respectively. The former regulates the propagation phase by using isotropic unit cells; the latter simultaneously manipulates the propagation and geometric phase to achieve the spin-decoupled phase control by arranging anisotropic unit cells. In addition, we also demonstrate the stable generation of a perfect Poincaré beam with arbitrary polarization and phase distribution on a hybrid-order Poincaré Sphere via a spin-decoupled metasurface, which is achieved by the linear superposition of two PVBs with orthogonal circular polarizations. The proposed scheme provides a compact and efficient platform for the generation and superposition of PVBs in THz region, and will speed up the progress of THz communication systems, complex light field generation, and quantum information sciences.

## 1. Introduction

It is well known that light has both spin angular momentum (SAM) and orbital angular momentum (OAM), which are manifested as circular polarization and azimuthal phase of a light beam, respectively [1,2]. Owing to the annular intensity distribution and the particular helical phase structure, light with OAM (namely vortex beams) has been widely used in optical communication [3], optical trapping [4], metrology [5], and quantum memories [6]. Terahertz (THz) waves located between far-infrared and microwave bands with rich frequency resources and have not been utilized extensively [7]. The related wireless THz communication has been recognized as one of the potential key technologies for high-speed communication in the future due to its great performance in high rate, broad bandwidth, and strong anti-interference [8,9]. The combination of THz waves and vortex beams will further extend communication capacity because different OAM modes are strictly orthogonal, and the multiple OAM states of vortex beams can be used to transmit different data [10,11]. However, the ring radiuses of these conventional vortex beams strongly depend on their topological charges, so it is not easy to combine multiple OAM beams into a single fiber used for multiplexed communications simultaneously [12,13].

A similar situation occurred in the vector vortex beams (VVBs) domain. Such spiral phase distribution and spatial inhomogeneous polarization state provide more interaction between light and matter, and have been utilized in high-resolution lithography [14], plasmonic couplers [15], and optical encryption [16]. Due to the inhomogeneous spatial polarization and phase distribution of VVBs, the Poincaré Sphere (PS) proposed to describe fundamental polarization states is not applicable [17]. Therefore, the higher-order Poincaré Sphere (HOPS) has been developed to visually describe the states of polarization and phase [18]. However, the HOPS model is still limited in specific cases because the OAM on the south and north poles is required to be opposite. In order to overcome this limitation, the hybrid-order Poincaré Sphere (HyOPS) has been proposed [19]. Similar to the PS, each point on a HyOPS describes a specific VVB represented by a linear superposition of the two poles with different phase and amplitude offsets. The beam on PS, HOPS, and HyOPS has been uniformly defined as the Poincaré beams (PBs), which describe most of polarized fields [20]. It has been proved that the superposition of vortex beams is an effective way to generate PBs [21]. However, this is usually complicated because the ring radius of conventional vortex beams is related to the topological charges, resulting in the polarization state being unstable and the intensity profile collapsing [22,23].

In order to overcome the above issues, Ostrovsky et al. introduced the concept of the perfect vortex beam (PVB), whose ring radius is independent of the topological charge [24,25]. Plenty of methods have been reported to obtain the PVB, including spatial light modulator [26], axicon [27], interferometer [28], and polymer-based phase plate [29]. Furthermore, the perfect Poincaré beam (PPB) theory that the ring radius is independent of the topological charge has been demonstrated using combinations of conventional optical components [30,31], such as axicons, spatial light modulators, q-plates, and Fourier transform lenses. However, these methods obtain the PVB and PPBs accompanied by complex optical path design, difficult fabrication, and bulky setup, running counter to the increasing miniaturization and integration requirements of photonic devices. Meanwhile, practical devices are absent in THz region, causing a challenge in generation and superposition of THz PVBs.

Metasurfaces, composed of subwavelength optical elements, can arbitrarily modulate light’s phase, polarization, and amplitude, providing an efficient, versatile platform for compact planar optics [32,33,34]. Despite many advances, the fixed wave-manipulation functionalities are still the main drawbacks in metasurfaces. In this case, many efforts have been devoted to achieving multifunctional and tunable metasurfaces that can dynamically control light upon external tuning. For example, active materials gain dynamic control of effective permeability, transmittance, and Fano resonances via the analytical designs of interfacial phase discontinuities [35,36,37]. Researchers have recently proposed a plasmonic metasurface in near-infrared (NIR) to generate PVBs based on geometric phase theory [38]. However, the transmission of the incident beam through the metasurface is deficient due to the large ohmic absorption caused by Au nanostructures. In 2021, Xie et al. designed a polarization-controllable dielectric metasurface in NIR to generate two different PVBs by switching the incident polarization utilizing propagation phase theory [39]. Compared to metal metasurface, dielectric metasurface displays higher work efficiency. In 2020, Bao et al. proposed a silicon metasurface to generate PPBs [40]. However, this design requires simultaneous phase and amplitude modulation as well as a complicated oblique incidence scheme, which limits its efficiency and operating bandwidth. In 2021, Zhou et al. demonstrated the efficient and broadband generation of PVBs at visible wavelengths using a single-layer dielectric metasurface [41]. However, the meta-device is single function, and the performance is sensitive to incident polarization due to the limitation of geometric phase theory, which will reduce its practicability. Meanwhile, a silicon metasurface has been designed to generate PBs in the THz range [42]. However, this work is only limited to the exceptional cases, such as radially or azimuthally polarized beams, and their annular intensity profiles are still dependent on the topological charge. Recently, Liu et al. proposed so-called quasi-perfect vortex beams (Q-PVBs), a new concept to realize vortex beams with a controllable divergence angle independent of topological charges. Moreover, the multichannel terahertz Q-PVBs generation is also theoretically and experimentally demonstrated [43]. However, this work is also limited by the geometric phase method. For example, it cannot be utilized to generate PVBs composed of arbitrarily circularly polarized vortex beams because there will always be one diverging circularly polarized output due to reversed phase sign. Therefore, to our best knowledge, a compact setup capable of generating an arbitrary PVB and further superposition to obtain PPBs in THz region is highly desirable and remains unexplored.

Here, two kinds of single-layer all-silicon metasurfaces have been designed to generate PVBs and PPBs in THz region. One generates polarization-independent PVBs by the propagation phase of cross-shaped unit cells. The other, made up of rectangular-shaped silicon pillars, can simultaneously employ the geometric and dynamic phase to generate spin-decoupled PVBs. Therefore, the radius and topological charges of the PVB can be conveniently varied by switching the incident spin states. Most importantly, two completely different PVBs can be mixed by a spin-decoupled metasurface. All orders of PPBs with different phase and polarization distributions can be generated by choosing the polarization states of incident light and designing topological charges of output PVBs. Thus, our work provides a potential compact platform for accelerating THz communication systems, complex light field generation, and quantum information sciences.

## 2. Materials and Methods

### 2.1. Polarization-Independent Metasurface

We design a metasurface that can generate polarization-independent PVBs. As shown in Figure 1a,b, the incident light with arbitrary polarization through the metasurface can produce a vortex beam such that radius is irrelevant to the topological charge. Theoretically, the complex amplitude of the PVB can be described as [24]: (1)$$E_{0}(\rho ,\theta )=\delta \left(\rho -\rho_{0}\right)\exp\left(il\theta \right)$$
where (ρ,θ)  represent the polar coordinates, δ(⋅) is the Dirac function in the polar coordinate system, *l* is the topological charge, and $\rho_{0\;}$ is the radius of bright annular intensity. Due to the Dirac function only existing in the ideal state, this perfect vortex model is not accessible in practice. According to the reported work, PVBs can be generated by taking the Fourier transform of an ideal higher-order Bessel beam [26], which can be presented in the following form,
(2)$$E_{B}\left(r,z\right)=J_{l}(k_{r}\rho )\exp\left(il\theta +ik_{z}z\right)$$Here, $J_{l\;}$ is the first kind of *l*-th order Bessel function; $k=\sqrt{K_{r}^{2}+K_{z}^{2}}=\frac{2\pi }{\lambda }$,$\;\left(k_{r},k_{z}\right)$ are the radial and longitudinal wave vectors, respectively, and λ is the incident wavelength. However, it is not easy to produce ideal Bessel beams experimentally. In most practical cases, the accessible Bessel model is the Bessel–Gaussian (BG) beam. Hence, the PVB beam is typically generated from the Fourier transform of the BG beam, which is usually achieved by a cascading optical setup, including a spiral phase plate, an axicon lens, and a Fourier transform lens [44]. Specifically, the generation of PVBs has two steps. First, a spiral phase plate and an axicon are used to convert the Gaussian beam to the corresponding BG beam. Second, a Fourier lens is utilized to transform the BG beam into PVBs. Due to the unprecedented light manipulating ability, a single-layer metasurface can work as a spiral phase plate, an axicon, and a Fourier transformation lens simultaneously, as shown in Figure 1c. The total phase profile can be expressed as:(3)$$\varphi_{total}\left(x,y\right)=\varphi_{\mathrm{spiral}}\left(x,y\right)+\varphi_{axicon}\left(x,y\right)+\varphi_{lens}\left(x,y\right)$$
(4)$$\varphi_{\mathrm{spiral}}\left(x,y\right)=l\cdot \arctan(\frac{x}{y})$$
(5)$$\varphi_{\mathrm{axicon}}\left(x,y\right)=-2\pi \frac{\sqrt{x^{2}+y^{2}}}{d}$$
(6)$$\varphi_{lens}\left(x,y\right)=\frac{-\pi (x^{2}+y^{2})}{\lambda f}$$
where *x* and *y* represent the position coordinates of the unit cell in the metasurface concerning the center of the origin, Equation (4) represents the spiral phase profile of an optical vortex with the topological charge *l*. Equation (5) shows an axicon phase profile, and *d* is the axicon period relating to the ring radius of the PVB. Equation (6) introduces the phase profile of the Fourier transform lens with the focal length of *f*, and *λ* is the working wavelength.

In order to realize the design requirement, we used the finite-difference time-domain (FDTD) method to optimize the geometric parameters of the unit cell. Single-layer plasmonic structures, depending on metallic materials with varying geometries, have been utilized to construct metasurfaces. However, they suffer from low efficiency due to the inherent ohmic loss and weak coupling between the incident and cross-polarized fields [45]. Although multilayer plasmonic structures can increase the efficiency, they also increase fabrication difficulties [46]. In addition, plasmonic structures are sensitive to the polarization of incident light, making them unsuitable for practical applications. It has been proved that the most straightforward approach to achieve polarization-independent performance is to find the unit cell with 4-fold symmetry [47]. Therefore, as shown in Figure 2a, the cross-shaped unit cell with the lattice constant of P = 130 μm, the height of H = 300 μm, and the exact width W = 50 µm are selected at the working frequency 0.75 THz. High-resistance silicon (*ρ* > 5000 Ω m) has been chosen as the designed material due to the negligible absorption and very low dissipation for terahertz wave as well as mature etching techniques, and its dielectric constant is set as ε = 11.9 [48,49].

By scanning the length (L) of the unit cell from 40 μm to 125 μm, both the transmittance and phase shift have been calculated, as shown in Figure 2b. Moreover, the inset illustrates the side views of the simulated magnetic amplitude distributions when L is fixed as 120 μm. The magnetic fields are mainly localized inside the silicon pillars, indicating that the unit cell can be seen as a truncated low-loss waveguide that works as a weakly coupled low-quality factor Fabry–Perot resonator [50]. Next, as shown in Figure 2c, eight units are chosen to realize the complete phase control with high transmittance. Their phase shifts cover the 2π range, and the transmittances are maintained over 70%. Figure 2d shows phase wavefronts under the x-polarized wave incident at the pillar height H of 330 μm and the length L of 40, 82, and 120 μm, respectively. Clearly, the relative phase difference between these pillars at the output port (at 330 μm) corresponds to π, and 2π, respectively.

### 2.2. Spin-Decoupled Metasurface

We design a spin-decoupled metasurface to generate two completely different PVBs only by varying the incident spin state. As shown in Figure 3a, left-circularly polarized (LCP) incident light through the metasurface can be transformed into a PVB with opposite-handedness, and the same metasurface can also transform right-circularly polarized (RCP) incident light into a different PVB with LCP state. Theoretically, the device must provide two independent spatial phase profiles corresponding to LCP and RCP, which cannot be achieved by using a single geometric phase or propagation phase.

For an anisotropic unit cell, the corresponding Jones matrix is:(7)$$T\left(x,y\right)=R\left(-\theta \right)TR\left(\theta \right)=\left(\begin{array}{cc} t_{x}\cos^{2}\theta +t_{y}\sin^{2}\theta & t_{y}\sin\theta \cos\theta -t_{x}\sin\theta \cos\theta \\ t_{y}\sin\theta \cos\theta -t_{x}\sin\theta \cos\theta & t_{x}\cos^{2}\theta +t_{y}\sin^{2}\theta \end{array}\right)$$
where $t_{x}=T_{x}e^{i\varphi_{x}},t_{y}=T_{y}e^{i\varphi_{y}}\;$are complex amplitude, $T_{i}$ and $\varphi_{i}$ represent the transmission amplitude and the phase delay while the incident light is along with the i-polarized wave with iϵ(x,y), respectively. $R\left(\theta \right)=\left[\begin{array}{cc} \cos\theta & -\sin\theta \\ \sin\theta & \cos\theta \end{array}\right]\;$is the rotation matrix with the rotation angle θ. When the metasurface is illuminated by circularly polarized light as $E_{in}=\frac{\sqrt{2}}{2}\left[\begin{array}{c} 1\\ \pm i \end{array}\right]$, the output electric field can be expressed as:(8)$$E_{out}=T(x,y)*E_{in}=\frac{\sqrt{2}}{4}\left(t_{x}+t_{y}\right)\left[\begin{array}{c} 1\\ \pm i \end{array}\right]+\frac{\sqrt{2}}{4}\left(t_{x}{-t}_{y}\right)e^{\pm i2\theta }\left[\begin{array}{c} 1\\ \mp i \end{array}\right]$$

Here$\;\left[\begin{array}{c} 1\\ i \end{array}\right]$ and $\left[\begin{array}{c} 1\\ -i \end{array}\right]\;$ represent LCP and RCP, respectively. The transmitted light can be divided into two parts: the co-polarized component without any phase regulation and the cross-polarized component with conjugated phase modulation ±2θ. Combining the orientation-dependent geometric phase with the dimension-dependent propagation phase effectively will address the spin-locked limitation of geometry metasurfaces [51,52]. Consider the unit cell is lossless ($T_{x}=T_{y}=1$) with π phase difference ($\left|\varphi_{x}\left(x,y\right)-\varphi_{y}\left(x,y\right)=\pi \right|)$, indicating the pillars working as a half waveplate. The output electric field can then be expressed as:(9)$$\left[\begin{array}{c} E_{out}^{x}\\ E_{out}^{y} \end{array}\right]=\frac{\sqrt{2}}{2}e^{\varphi_{x}}e^{\pm i2\theta }\left[\begin{array}{c} 1\\ \mp i \end{array}\right]$$

Assuming the expected phase distributions of the spin-decoupled are $\varphi_{1}\left(x,y\right)\;$and $\varphi_{2}\left(x,y\right)$. From the Equation (9), we can derive that:
(10)$$\{\begin{array}{c} \varphi_{x}(x,y)+2\theta (x,y)=\varphi_{1}(x,y)\\ \varphi_{x}(x,y)-2\theta (x,y)=\varphi_{2}(x,y) \end{array}$$
where $\varphi_{x}\left(x,y\right)\;$represents the propagation phase, which is influenced by the material and geometry parameters of the unit pillars; ±2θ(x,y) is the geometric phase, which is only determined by the rotation angle of the unit cell. Therefore, the expected phase distributions $\varphi_{1}\left(x,y\right)\;$and $\varphi_{2}\left(x,y\right)$ can be satisfied simultaneously by designing the geometry and rotation angle of the unit structure, as shown in Figure 3b.

To construct the metasurface, the sub-wavelength rectangular-shaped silicon pillar has been chosen, as shown in Figure 4a. The simulation is based on the FDTD method to obtain the pillars’ transmittance and phase delay database under the x- and y-polarized incident waves. The length L and width W varied from 35 μm to 135 μm with the fixed lattice constant of P = 140 μm and the height of H = 330 μm at the working frequency of 0.75 THz. Figure 4b,c show the transmittance as the function of the pillar sizes in L and W, respectively. The corresponding simulated phase shifts ($\varphi_{x},\;\varphi_{y}$) are illustrated in Figure 4d,e. Both $\varphi_{x\;}$ and $\varphi_{y}$ cover 2π degrees in the scanning range, and all the transmittances are more than 70%. Therefore, by carefully selecting L and W from the database, a nearly arbitrary phase combination ($\varphi_{x},\;\varphi_{y}$) with high transmittance can be obtained for the silicon pillar. For our design, nine different silicon pillars are chosen, and the corresponding transmittances, phase shifts, and polarization conversion efficiencies (PCE, defined as the intensity ratio between the converted spin component and the total transmitted beam) are shown in Figure 4f. Phase shifts $\varphi_{x}$ (red square) and $\varphi_{y}\;$(green square) are linearly increased with a step of 2π/9. The corresponding transmittance $T_{x}$ (red sphere) and $T_{y}$ (green sphere) are more than 70%. Moreover, the PCE (orange column) is maintained above 70%. Most importantly, the phase differences between $\varphi_{x}$ and $\varphi_{y}$ are close to π, and the transmittance $T_{x}$ and $T_{y}$ are almost equal, indicating these selected pillars are working well as the half-wave plates at 0.75 THz. The specific geometric parameters and PCE of the nine selected silicon pillars are listed in Table 1.

### 2.3. Superpositions of PVBs by a Spin-Decoupled Metasurface

Under the paraxial approximation, the vectorial field of a monochromatic Poincaré beam can be decomposed into the superposition of two orthogonal circularly polarized vortex beams with different coefficients $\psi_{N}^{l}$ and $\psi_{s}^{m}$ as described [22]:(11)$$|\psi_{l,m}\rangle =\psi_{N}^{l}|R_{l}\rangle +\psi_{S}^{m}|L_{m}\rangle$$
where $|R_{l}\rangle =e^{il\varphi }\left(e_{x}+ie_{y}\right)/\sqrt{2}$
, $|L_{m}=e^{im\varphi }\left(e_{x}-ie_{y}\right)/\sqrt{2}$ are the RCP and LCP vortex beams with topological charges of l and *m*, respectively, while ***e_x_*** and ***e_y_*** are the unit vectors along the *x* and *y* axes. φ is the azimuthal angle in the polar coordinates. In order to obtain PPBs that radius is irrelevant to topological charges, Equation (11) can be further deduced as:(12)$$|\psi_{l,m}\rangle =\psi_{N}^{l}|POV_{R},l\rangle +\psi_{S}^{m}|POV_{L},m\rangle$$Here, $|POV_{R},l=e^{i\varphi_{total}}\left(e_{x}+ie_{y}\right)/\sqrt{2}$and$|POV_{L},m=e^{i\varphi_{total}}\left(e_{x}-ie_{y}\right)/\sqrt{2}$ represents the RCP and LCP PVBs with topological charges of l and *m*, $\varphi_{total}\;$represent the phase profile of PVBs obtained from Equation (3).

According to Equation (12), two essential parts, orthogonal circularly polarized PVBs and coefficients $\psi_{N\;}^{l}$ and $\psi_{s}^{m}$, should be resolved for generating arbitrary PPBs. In order to describe the process more clearly, we use the PS to describe scalar fields of incident light and the HyOPS to describe the vectorial field of output PPBs. As shown in Figure 5a, all the possible states of fundamental polarization can be mapped onto the PS. Because the polarization state of the scalar field can be represented with orthogonal circular polarization, the coefficients $\psi_{N}^{l}$ and $\psi_{s}^{m}$ vary with the incident polarization. As shown in Figure 5b, the spin-decoupled metasurface can generate two completely different PVBs with orthogonal circularly polarization. If the axicon period and the focal length are identical, arbitrary PPBs can be generated by superposing the two circularly polarized PVBs on a HyOPS. As shown in Figure 5c, each point on the surface of HyOPS with spherical coordinates (α,β) can indicate a PPB, where α∈(0,2π), β∈(0,π). The two poles of the HyOPS represent two PVBs with opposite circular polarization and topological charges of l and m, respectively. Points on the equator of HyOPS represent a superposition of equal intensities and different phase differences of the opposite circular polarization states. For example, the points (0,π/2), (π,π/2) represent the radially and azimuthally vector beam, respectively.

For the PPBs on a HyOPS, the polarization state can be mapped by representing the Stokes parameters in the spherical Cartesian coordinates, which are defined as follows [20]:(13)$$S_{0}^{l,m}={\left|\psi_{N}^{l}\right|}^{2}+{\left|\psi_{S}^{m}\right|}^{2}$$
(14)$$S_{1}^{l,m}=2\left|\psi_{N}^{l}\right|{\left|\psi_{S}^{m}\right|}^{2}\cos\Phi$$
(15)$$S_{2}^{l,m}=2\left|\psi_{N}^{l}\right|{\left|\psi_{S}^{m}\right|}^{2}\sin\Phi$$
(16)$$S_{3}^{l,m}={\left|\psi_{N}^{l}\right|}^{2}-{\left|\psi_{S}^{m}\right|}^{2}$$
Here $\Phi =\arg\left(\psi_{N}^{l})\right)-\arg(\psi_{N}^{l})$ is the phase difference between the two PVBs. ${\left|\psi_{N}^{l}\right|}^{2}$and ${\left|\psi_{S}^{m}\right|}^{2}$ are the intensity of RCP and LCP PVB, respectively. The concepts of polarization order P=(l−m)/2 and the topological Pancharatnam charge $l_{P}=\left(l+m\right)/2$ are also used to characterize PPBs [22]. This can be explained by deducting Equation (12) as:(17)$$|\psi_{l,m}\rangle =e^{il_{p}\varphi }\left(\psi_{N}^{l}e^{ip\varphi }|POV_{R},l\rangle +\psi_{S}^{m}e^{-ip\varphi }|POV_{L},m\rangle \right)$$Equation (17) indicates that the polarization state of PPBs is only determined by the parameters within the brackets, that is, the polarization order P and the HyOPS coordinate(α,β). The topological Pancharatnam charge $l_{P}$ characterizes the phase distribution of the PPBs, which can be represented as $e^{i\varphi l_{P}}$.

## 3. Results and Discussion

Based on the above design principles, we designed and numerically simulated several metasurface devices (diameter is 14 mm) to demonstrate the practicability of our proposed design by employing the FDTD method. The perfectly matching layers (PMLs) were used in the x, y, and z directions. Plane-wave sources were used in all simulations.

### 3.1. Polarization-Independent PVBs Generator 

The generation of polarization-independent PVBs has been investigated. The design phase distribution has been described in Equation (3). Figure 6a displays the intensity profiles in the x-z plane of the generated PVB with the parameters of d = 4 mm and l = 2 at 0.75 THz. Figure 6b plots the corresponding cross-section images of the generated PVB at different longitudinal positions. The THz beam passing through the metasurface presents an annular ring profile near the focal plane (z = 14 mm). Moreover, the intensity profile of the ring maintains a circular shape, but the radius increases with increasing propagation distance.

Figure 7a depicts the normalized intensity and phase distribution of the generated PVBs with different topological charges from l = 1 to l = 4 at the focus point when d = 3 mm and 4 mm, respectively. The focal length of the Fourier- transformed lens is designed as f = 14 mm at 0.75 THz. Figure 7b shows the corresponding cross-sections of the normalized intensity profiles extracted along the x-direction (solid line) and y-direction (dashed line). The consistent cross-sectional intensity profiles show that the generated vortex beams are insensitive to the topological charge, and the phase distribution is consistent with the designed topological charge. The full-wave simulation generation efficiencies (defined as the ratio of the PVB’s intensity to the intensity of the incident beam) of all polarization-independent metasurfaces are more than 49%. The polarization-independent properties and the relationship between the ring radius and the axicon period are analyzed in Appendix A. Appendix A characterizes the working bandwidth of the polarization-independent PVBs generator.

### 3.2. Spin-Decoupled PVBs Generator

According to theoretical analysis of a spin-decoupled PVBs generator, a single metasurface can generate two distinct PVBs. The corresponding phase distribution ($\varphi_{1},\;\varphi_{2}$) can be expressed as follows:(18)$$\{\begin{array}{c} \varphi_{1}(x,y)=l_{1}\cdot \arctan(\frac{x}{y})+(-2\pi \frac{\sqrt{x^{2}-y^{2}}}{d_{1}})+\frac{-\pi (x^{2}+y^{2})}{\lambda f_{1}}\\ \varphi_{2}(x,y)==l_{2}\cdot \arctan(\frac{x}{y})+(-2\pi \frac{\sqrt{x^{2}-y^{2}}}{d_{2}})+\frac{-\pi (x^{2}+y^{2})}{\lambda f_{2}} \end{array}$$
where *λ* is the incident wavelength, *l*_*i*_, *f*_*i*_, and *d*_*i*_ represent corresponding parameters for the incident RCP (*i* = 1) and LCP (*i* = 2) light, respectively. Two metasurface devices are designed based on Equations (10) and (18) to demonstrate the above function. The first metasurface is designed with *l*_1_ = 2, *l*_2_ = 3, *d*_1_ = *d*_2_ = 4 mm, and *f*_1_ = *f*_2_ = 14 mm at 0.75 THz. As shown in Figure 8a,c, when the incident light is converted from RCP to LCP, the transmitted light is converted into a corresponding cross-polarized component, and the ring radius does not change significantly, but the topological charge *l* changes from 2 to 3, which is consistent with our design. In addition, we can also control the ring radius of generated PVBs by switching the incident spin state. The second device with the parameters *l*_1_ = *l*_2_ = 3, *d*_1_ = 4 mm, *d*_2_ = 6 mm, and *f*_1_ = *f*_2_ =14 mm at 0.75 THz is also designed. As shown in Figure 8b,d, the topological charge value l does not change, but the ring radius is inversely proportional to the axicon periods d when the incident light is converted from RCP to LCP. It is worth noting that the transmitted light still carries with a co-polarized component, as shown in Figure 8a,b, because the selected unit cells are not a perfect half-wave plate. The full-wave simulation generation efficiencies of the spin-decoupled metasurfaces were more than 53%, which can be improved by selecting materials with higher transmittance and a more careful phase scanning operation with smaller step size (more details on working bandwidth of the meta-device seen also in Appendix A). Therefore, the spin-decoupled and polarization-independent PVB meta-devices will pave the way for developing a miniaturized and convenient platform for OAM-related applications in the terahertz band, especially in improving the transmission capacity of terahertz communications.

### 3.3. Superpositions of Perfect Vortex Beams

According to the theoretical analysis of generating PPBs in Section 2.3, the topological charges (l, m) of orthogonal circularly polarized PVBs and the corresponding coefficients $\psi_{N}^{l}$ and $\psi_{s}^{m}$ can determine a PPB. Specifically, different topological charges (l, m) satisfied by spin-decoupled metasurface will determine a unique HyOPS, different coefficients $\psi_{N}^{l}$ and $\psi_{s}^{m}\;$will define the position of a PPB on this HyOPS by calculating Stokes parameters. In order to verify the above principle, three spin-decoupled metasurfaces with the expected parameters d = 4 mm, and f =14 mm at 0.75 THz have been designed. The relevant polarization order P and topological Pancharatnam charge $l_{P}$ are $P_{1}=\left(l_{1}-m_{1}\right)/2=-1$, $l_{P1}=\left(l_{1}+m_{1}\right)/2$=0, $P_{2}=\left(l_{2}-m_{2}\right)/2=-2$, $l_{P2}=\left(l_{2}+m_{2}\right)/2\;$= 0, and $P_{3}=\left(l_{3}-m_{3}\right)/2=-1$, $l_{P3}=\left(l_{3}+m_{3}\right)/2=2$. Five points with their coordinates on the HyOPS are selected, given by the red dots in Figure 9a, and the corresponding incident polarization states are represented by the red points located on the PS shown in Figure 9b. Figure 9c shows the intensity patterns of three designed metasurfaces in the x-y plane at the focus point. The annular intensity patterns are obtained by a horizontal linear polarizer to analyze the polarization property. Because the polarization order P determines the number of polarization rotations per round trip, the transmitted intensity patterns through a horizontal linear polarizer will have 2|P| lobes.

The case with the same topological Pancharatnam charge $l_{P}$ and different polarization order P is firstly discussed. As shown in the first row and the second row in Figure 9c, the transmitted intensity patterns are split into two and four lobes, respectively, which satisfy the theoretical predictions of $P_{1}=-1$, $P_{2}=-2$. Then, the influence of the topological Pancharatnam charge $l_{P}$ is considered. The polarization order is fixed as P=−1, and varied with the topological Pancharatnam charge $l_{P}=0,\;2$, respectively. The simulated annular intensity distributions in the first and third row show the same annular patterns, demonstrating the independence of the polarization state of PPBs from the topological Pancharatnam charge, matching well with the theoretical prediction of Equation (17). It should be noticed that the third row has a slight rotation of the measured intensity induced by the Gouy phase [53]. In principle, the Gouy phase is dependent on the topological charge, and the rotation of the intensity distribution around the beam axis occurs only when the topological charges |l|≠|m|. Therefore, this approach provides a potential solution to measure the Gouy phase.

To further characterize the capability of arbitrary PPBs generation, another five incident lights represented by the blue points on the Poincaré sphere are chosen, as shown in Figure 9b, where the two circular eigenstates have a phase difference of π/2. The corresponding PPBs with their coordinates on the HyOPS are expressed by the blue dots in Figure 9a, and the measured intensity patterns through a horizontal linear polarizer are shown in Figure 9d. Unsurprisingly, the introduced phase difference of π/2 between the two circular eigenstates results in the change of coefficients $\psi_{N}^{l}$ and $\psi_{s}^{m}$, further causing the rotation of the intensity profiles and demonstrating the feasibility of PPBs by changing the polarization of incident light.

In addition, the Stokes parameters have also been used to verify inhomogeneous polarization states of the PPBs, which are given by [54]: 

(19)$$S_{0}=I_{0}+I_{90}$$(20)$$S_{1}=I_{0}-I_{90}$$(21)$$S_{2}=I_{45}-I_{135}$$(22)$$S_{3}=I_{R}-I_{L}$$
where $I_{0}$, $I_{90}$, $I_{45}$ and $I_{135}$ are the intensities of PPBs through a linear polarizer oriented at 0°, 45°, 90°, and 145°. $I_{R}$ and $I_{L}$ are the intensities of PPBs through a right and a left circular polarizer, which can be achieved by cascading a linear polarizer and a quarter waveplate. With these Stokes parameters, the detailed polarization of a PPB can be obtained. The spherical coordinates (α,β) of the points on the HyOPS can be determined by the Stokes parameters:



(23)
$$\alpha =arctan\left(S_{2}/S_{1}\right)$$


(24)
$$\alpha =arccos\left(S_{2}/S_{1}\right)$$



Figure 10a shows the calculated results of normalized Stokes parameters by choosing point III in Figure 9a for three kinds of PPBs. The first Stokes parameter *S*_0_, representing the light intensity, shows the same ring pattern consistent with our theoretical analysis. By analyzing Stokes parameters, *S*_1_ and *S*_2_, the polarization orientation angle σ can be determined by:(25)$$\sigma =\frac{1}{2}\arctan(\frac{S_{2}}{S_{1}})$$

The corresponding polarization orientations (orange double arrows) and distributions of the PPBs are shown in Figure 10b. It can be seen that the polarization orientations rotate 2π and 4π per round trip for cases of $P_{1}=P_{3}=-1$, $P_{2}=-2$, respectively, which is consistent to the definition of polarization order. It should be noticed that the spherical coordinate of point III is (0,π/2), which means the Stokes parameter S_3_ is zero theoretically. The error appearing in Figure 10a mainly comes from two reasons: the phase difference of the metasurface between the actual phase and the theoretical phase, and the act that the selected unit cells are not a perfect half-wave plate. The full-wave simulation generation efficiencies of the PPBs (defined as the ratio of the PPB’s intensity to the intensity of the incident beam) were more than 51%. These results and the analysis demonstrate that spin-decoupled metasurfaces can efficiently generate arbitrary PPBs on the HyOPS in THz region. Therefore, the terahertz PVB meta-devices designed here will also provide a potential compact platform for vector beam related applications in the terahertz band, such as reconstructing incident photons’ frequency and polarization state [55].

## 4. Conclusions

In conclusion, we propose a general method to generate PVBs using a single-layer dielectric metasurface in THz regime. This is achieved by integrating the function of spiral phase plate, axicon, and Fourier transformation lens into a single-layer metasurface. The ring radiuses of these generated vortex beams are independent of the topological charges and inversely proportional to the axicon periods. Meanwhile, the unit-cell cross-section geometries with 4-fold symmetry can make the generated PVBs independent of incident polarization. Moreover, to realize more freedom, a spin-decoupled metasurface is designed by simultaneously manipulating the dynamic phase and geometric phase of the unit cells. The metasurface can generate spin-decoupled PVBs with arbitrary radius or topological charges by changing the incident spin states. In addition, the generation of an arbitrary perfect Poincaré beam at any position on the surface of the HyOPS is also demonstrated by superposing two perfect vortex beams. Various perfect Poincaré beams were generated, and the results agreed well with the theory. This work will provide more opportunities and possibilities for THz communication, complex light field generation, and quantum information sciences.

## Figures and Tables

**Figure 1 nanomaterials-12-03010-f001:**
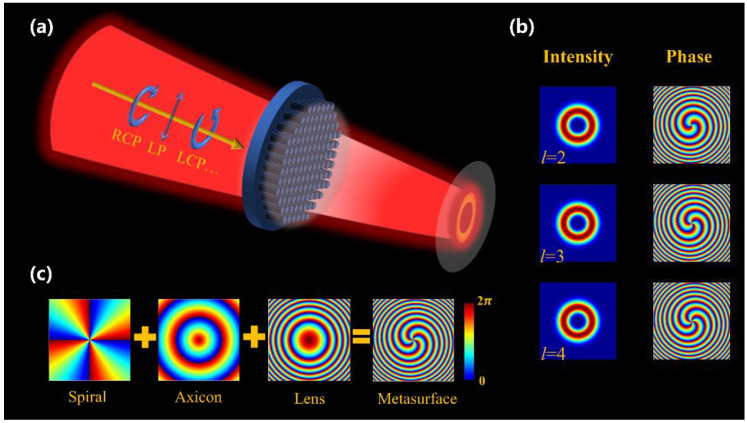
Generation of polarization-independent PVBs. (**a**) Schematic of the polarization-independent metasurface. (**b**) Simulated intensity distribution and phase distribution of PVBs. (**c**) The phase profile of the designed metasurface includes the phase of a spiral phase plate, an axicon, and a Fourier transformation lens.

**Figure 2 nanomaterials-12-03010-f002:**
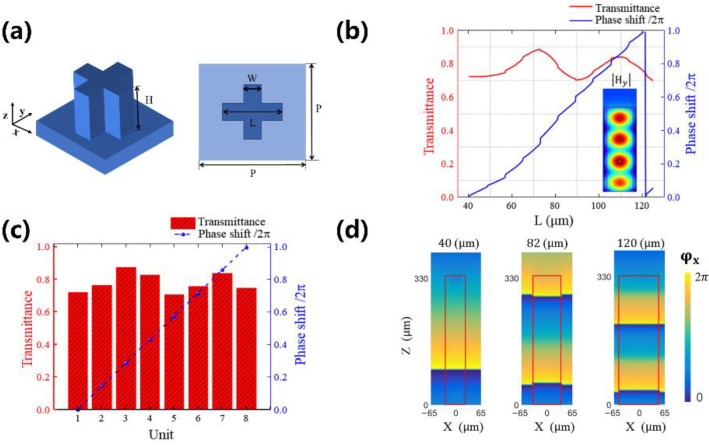
Characterization of the designed unit cell for modulating the propagation phase. (**a**) Schematic of the unit cell: the lattice constant P, the unit length L, the width W, and the height H. (**b**) Transmittance and phase shifts as a function of the unit length L. The inset illustrates the side views of the simulated magnetic amplitude distributions of the unit cell with L = 120 μm. (**c**) The phase shifts and transmittance of eight selected silicon units. (**d**) Simulated phase wavefront in three silicon units with L = 40, 82, and 120 μm, respectively.

**Figure 3 nanomaterials-12-03010-f003:**
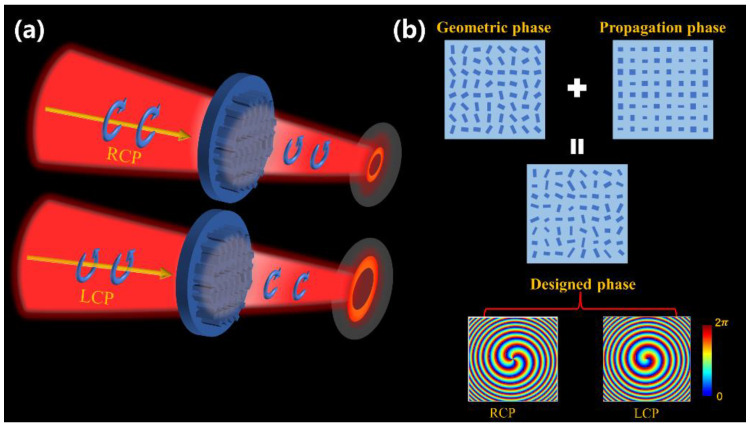
Generation of spin-decoupled PVBs. (**a**) Schematic of the spin-decoupled metasurface. (**b**) The designed total phase of the single-layer metasurface simultaneously contains the geometric phase and the dynamic phase, which are only related to geometry parameters and the rotation angle of the unit structure, respectively.

**Figure 4 nanomaterials-12-03010-f004:**
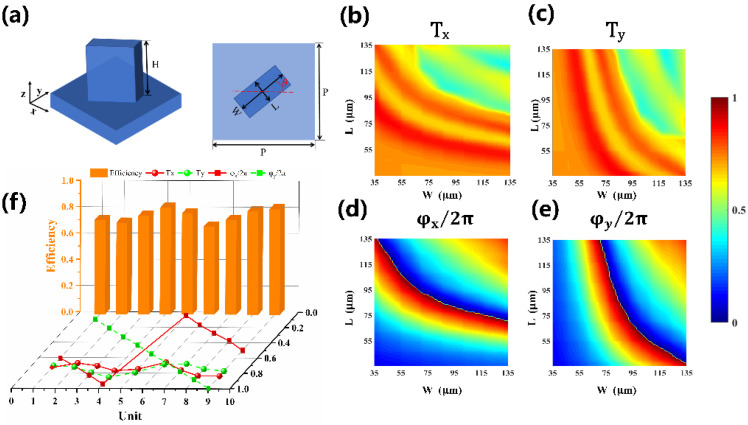
(**a**) Schematic of the rectangular-shape silicon pillar: the lattice constant P, the length L, the width W, and the height H, respectively. The angle θ represents the rotation of the unit cell. (**b**–**e**) Simulated transmittance and phase shift as a pillar sizes (L, W) function under x-polarized and y-polarized illumination at 0.75 THz, respectively. (**f**) Transmittances, phase shifts, and polarization conversion efficiencies of the nine selected unit cells.

**Figure 5 nanomaterials-12-03010-f005:**
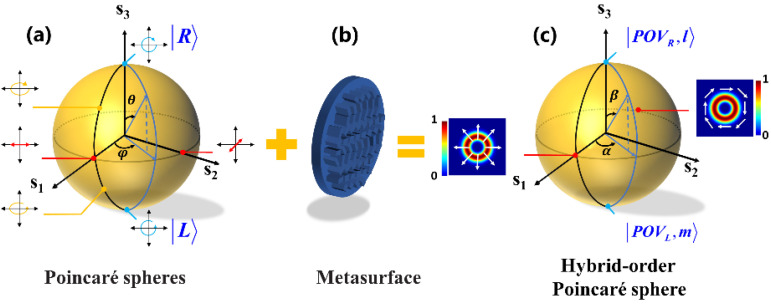
Illustration of generating PPBs through a dielectric metasurface with the aid of PS and HyOPS. (**a**) Schematic of Poincaré sphere. A point on the Poincaré sphere represents the polarization state of the incident light. (**b**) Schematic of the spin-decoupled metasurface that can superpose two completely different PVBs. (**c**) The output PPBs is represented by points on the HyOPS. The inset illustrates the radially and azimuthally polarized beams’ intensity and polarization distribution (depicted by white arrows) on the HyOPS with *l* = −1 and *m* = 1.

**Figure 6 nanomaterials-12-03010-f006:**
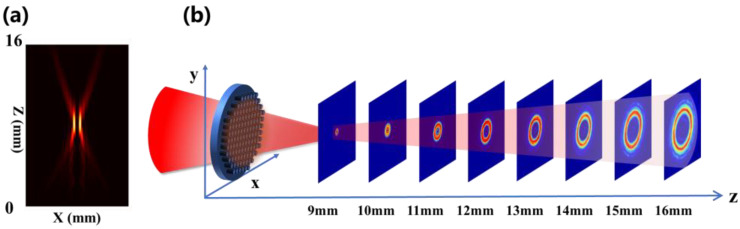
(**a**) The intensity profiles in the x-z plane of the generated PVB with the parameters of d = 4 mm and l = 2. (**b**) Corresponding 2D cross-section images of the generated PVB at different longitudinal positions.

**Figure 7 nanomaterials-12-03010-f007:**
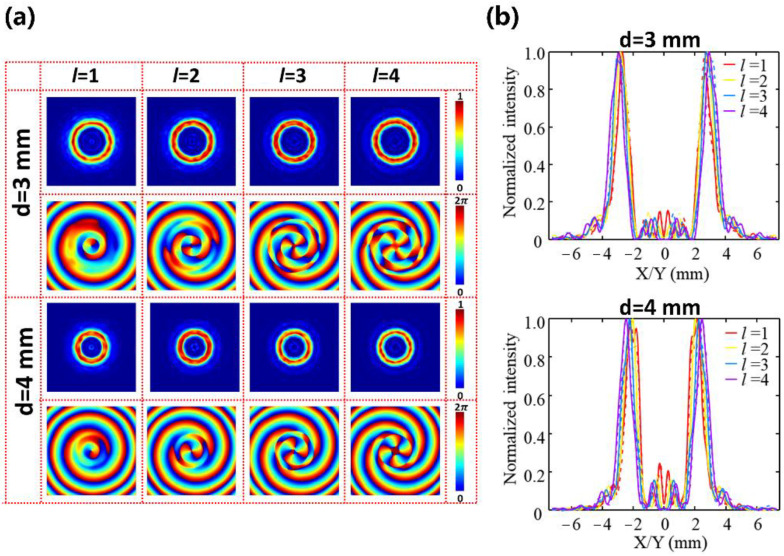
(**a**) The intensity and phase profiles of the PVBs with the topological charges varied from l = 1 to l = 4 when the axicon periods d were selected as d = 3 mm and 4 mm, respectively. (**b**) Corresponding normalized cross-section of the annular intensity distribution along x-direction (solid line) and y-direction (dashed line) at 0.75 THz when d = 3 mm (upper) and d = 4 mm (down), respectively.

**Figure 8 nanomaterials-12-03010-f008:**
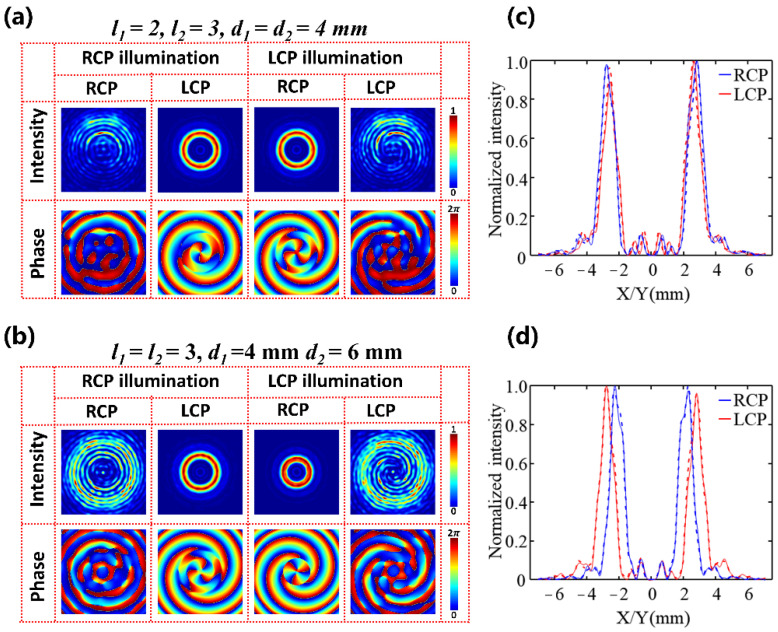
(**a**,**b**) The normalized intensity profile and phase distribution for devices I and II at focus point when only the RCP or LCP beam is incident. (**c**,**d**) Corresponding normalized cross-section of the annular intensity distribution for RCP (blue) and LCP (red) polarization along x-direction (solid line) and y-direction (dashed line).

**Figure 9 nanomaterials-12-03010-f009:**
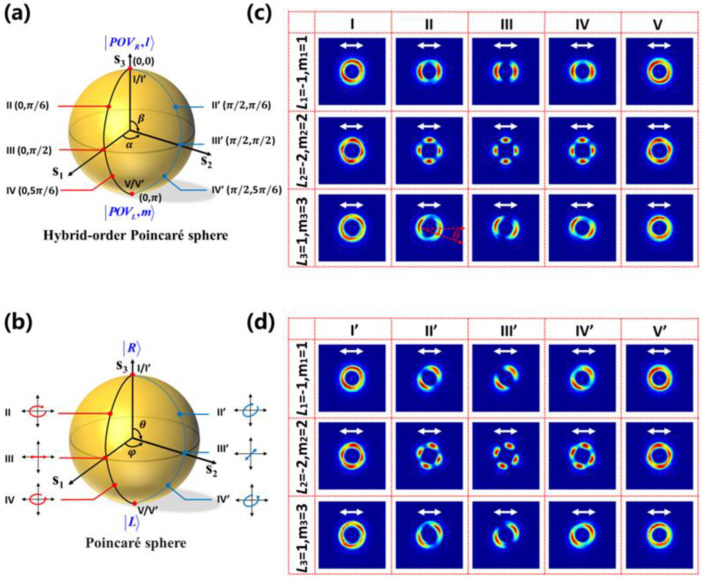
Characterization of the generated PPBs. (**a**) The selected states of PPBs are represented by red and blue points on the HyOPS with their coordinates. The phase difference between the two circular eigenstates of the selected blue dots is π/2. (**b**) Corresponding incident polarization states are represented by the red and blue points on the Poincaré sphere. (**c**,**d**) The simulated intensity profiles of the output PPBs correspond to the red points (**c**) and blue points (**d**) in (**a**) for three designed metasurfaces through a horizontal linear polarizer (depicted by the white double arrow).

**Figure 10 nanomaterials-12-03010-f010:**
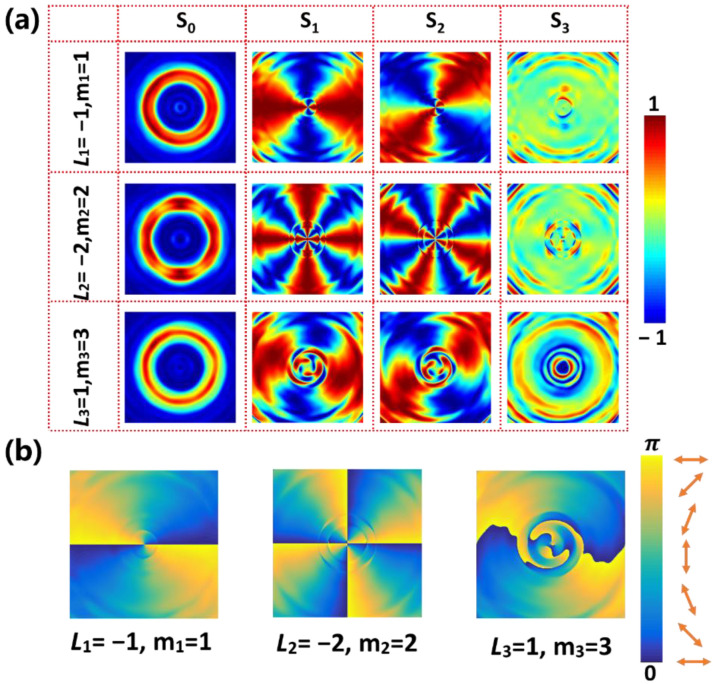
(**a**) The calculated normalized Stokes parameters correspond to point III in Figure 9a for three kinds of PPBs. (**b**) The calculated polarization orientations and distributions for three PPBs. Note that the orange double arrow represents the linear polarization orientation.

**Table 1 nanomaterials-12-03010-t001:** Geometric parameters and polarization conversion efficiency for the selected unit cells.

Unit	1	2	3	4	5	6	7	8	9
L (μm)	109	39	49	55	60	107	107	103	103
W (μm)	60	98	101	105	105	35	44	51	58
PCE (%)	73.8	70.4	75.1	80.1	77.0	70.6	71.9	78.6	80.3

## Data Availability

Data sharing is not applicable to this article.

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
