# Peer review of "Generation and Superposition of Perfect Vortex Beams in Terahertz Region via Single-Layer All-Dielectric Metasurface"

_nanomaterials, 2022, doi:10.3390/nano12173010_

Round 1

Reviewer 1 Report

The authors proposed a method to generate perfect vortex beams (PVBs) using a single-layer dielectric metasurface in THz regime. In addition, they demonstrated the generation of an arbitrary perfect Poincare beam (PPB) at any position on the surface of the HyOPS by superposing two PVBs. The method is promising while the results are interesting. I will recommend the publication of this manuscript in Nanomaterials if the authors can address the following comments convincingly.

1. The novelty of the proposed methods to generate PVBs and PPBs is not clear. Compared with the methods shown in Nanophotonics, 2020, 9, 3393-3402 and Nature Communications, 12, 2230 (2021), whether there is anything new in the proposed methods should be discussed.

2. High-resistance silicon is used as the material for designing and simulating the THz metasurfaces in this manuscript. Its detailed optical parameters including the real and imaginary parts of the dielectric constant and the dispersion in the THz range should be given in the manuscript or SI.

3. THz is a key point for the metasurface design in this manuscript. Whether there are any other excellent materials for designing metasurfaces in the THz range should be discussed.

4. The caption of Figure 10 (a) is incorrect. The calculated Stokes parameters should correspond to point III in Figure 9(a) rather than Figure 10(a).

Reviewer 2 Report

The authors proposed two perfect vortex beam (PVB) generators employing all-dielectric metasurface to achieve polarization-independent PVB and spin multiplexed PVB. The scheme of the meta-device is original, and the mechanism of its operation has been described with due care. Nevertheless, there are a few points that require additional discussion:

1.      In the introduction, the authors omitted several functionalities of THz metamaterials, please see:  Physical Review B 79.24 (2009): 241108; Applied physics letters 106.9 (2015): 092905; Scientific reports 7.1 (2017): 1-8, etc.

2.      Why was this shape of the meta-surface created? What is its greatest uniqueness? It is true that the authors wrote a few words about the meta-surface, but this translation is far from sufficient. Please justify it by comparing the developed model with other similar publications.

3.      The metamaterial resonance seen in Figure 2b is quite weak. What steps can be taken to optimize it and what impact this will have on PVB generation? Please explain.

4.      There is no detailed description of the applications of the proposed meta-device. In what systems can their photonic properties be used?

Round 2

Reviewer 1 Report

I recommend the publication of this manuscript in Nanomaterials.